# Stem cell heterogeneity drives the parasitic life cycle of *Schistosoma mansoni*

Bo Wang[1,2,3]*, Jayhun Lee[3†‡], Pengyang Li[1†], Amir Saberi[3†§], Huiying Yang[1], Chang Liu[4], Minglei Zhao[4], Phillip A Newmark[3‡]*

[1]Department of Bioengineering, Stanford University, Stanford, United States; [2]Department of Developmental Biology, Stanford University School of Medicine, Stanford, United States; [3]Department of Cell and Developmental Biology, Howard Hughes Medical Institute, University of Illinois at Urbana-Champaign, Urbana, United States; [4]Department of Biochemistry and Molecular Biology, University of Chicago, Chicago, United States

*For correspondence:
wangbo@stanford.edu (BW);
pnewmark@morgridge.org (PAN)

†These authors contributed equally to this work

Present address: ‡Department of Integrative Biology, Howard Hughes Medical Institute, Morgridge Institute for Research, University of Wisconsin–Madison, Madison, United States; §Division of Cardiology, Johns Hopkins University School of Medicine, Baltimore, United States

**Abstract** Schistosomes are parasitic flatworms infecting hundreds of millions of people. These parasites alternate between asexual reproduction in molluscan hosts and sexual reproduction in mammalian hosts; short-lived, water-borne stages infect each host. Thriving in such disparate environments requires remarkable developmental plasticity, manifested by five body plans deployed throughout the parasite's life cycle. Stem cells in *Schistosoma mansoni* provide a potential source for such plasticity; however, the relationship between stem cells from different life-cycle stages remains unclear, as does the origin of the germline, required for sexual reproduction. Here, we show that subsets of larvally derived stem cells are likely sources of adult stem cells and the germline. We also identify a novel gene that serves as the earliest marker for the schistosome germline, which emerges inside the mammalian host and is ultimately responsible for disease pathology. This work reveals the stem cell heterogeneity driving the propagation of the schistosome life cycle.
DOI: https://doi.org/10.7554/eLife.35449.001

## Introduction

Flatworms include more than 44,000 parasitic species that form one of the largest groups of metazoan endoparasites (*Loker and Hofkin, 2015*). Their life cycles typically involve asexually and sexually reproducing stages, each with its own distinct body plan and strategy to enhance transmission between multiple hosts (*Clark, 1974*; *Pearce and MacDonald, 2002*; *Viney and Cable, 2011*). Although the life cycles of these parasites were established more than a century ago, they have only recently been studied in cellular and molecular terms (*Matthews, 2011*). Since many parasitic flatworms are pathogenic, their life cycles are also the routes for disease transmission (*Hoffmann et al., 2014*). Therefore, a deeper understanding of these life cycles is significant from both basic science and medical perspectives, as blocking transmission is an effective approach to fighting parasitic diseases.

Focusing on the cells that may drive such parasitic life cycles, we study *Schistosoma*, a parasitic flatworm infecting over 250 million people, which causes the major neglected tropical disease, schistosomiasis (*Hoffmann et al., 2014*). Schistosomes are transmitted through snail intermediate and human definitive hosts. Their life cycle begins with the parasite egg excreted from the mammalian host into water, releasing a free-swimming miracidium larva. The miracidium penetrates a snail host and transforms into a mother sporocyst that undergoes asexual clonal expansion to produce many daughter sporocysts that leave the mother and colonize other snail tissues. These daughters either self-renew to produce more daughters or enter embryogenesis to produce infective cercariae

**eLife digest** Parasitic flatworms called schistosomes infect around 250 million people, causing the disease schistosomiasis. Schistosomes live complex lives, spending part of their life cycle inside snails and part of it inside mammals; short-lived, water-borne stages infect each of these hosts. To thrive in such different environments, schistosomes go through several life-cycle stages. At each stage the flatworms transition to a new body plan adapted to its new environment. Understanding how these transitions occur could help researchers devise new strategies for eliminating these parasites.

Previous research suggested that stem cells help schistosomes transition to new body plans. Stem cells have the ability to transform into many different cell types, and have been found in schistosome larvae and adults. However, the relationship between the larval and adult stem cells was not clear.

Wang et al. used transcriptional profiling, a technique that measures the genes currently in use in different cells, to study the stem cells in the schistosome species *Schistosoma mansoni*. This uncovered four types of stem cell, each of which uses a slightly different combination of genes. Examining the behaviour of these cells at different schistosome life-cycle stages revealed that certain larval stem cells produce adult stem cells. Other larval stem cells seem to be the source of the 'germline' cells that make gametes (egg and sperm) and allow the parasites to reproduce sexually.

Schistosomes only produce germline cells when they are inside mammals. Wang et al. found that as juvenile flatworms develop inside mouse blood vessels, a gene called *eledh* becomes active in some of their stem cells. Further investigation showed that this activity is the earliest indicator that germline cells are developing and is also required for proper development of the germline. This knowledge, along with future work to characterize the roles of the stem cell populations identified by Wang et al., could ultimately help researchers develop new ways to stop the spread of schistosomiasis.

DOI: https://doi.org/10.7554/eLife.35449.002

(*Cheng and Bier, 1972*; *Jourdane and Theron, 1980*; *Whitfield and Evans, 1983*; *Taft et al., 2009*). This cloning process is repeated, allowing massive numbers of cercariae to be produced from a single miracidium. Mature cercariae emerge from the snail into water, then burrow through the skin of a mammalian host to become schistosomula. This transition initiates the sexual portion of the life cycle. Schistosomula then migrate to species-specific niches in the host vasculature and develop into juvenile worms (*Basch, 1991*; *Wilson, 2009*). Juveniles remodel their tissues extensively to build a functional digestive system, and after they begin feeding on host blood, undergo massive growth and develop sexual reproductive organs de novo (*Clegg, 1965*). Male and female worms pair to produce fertilized eggs, which are excreted to continue the life cycle.

A long-standing hypothesis proposes that a lineage of totipotent stem cells, called 'germinal cells', persists throughout the schistosome life cycle and drives reproduction (*Cort et al., 1954*; *Clark, 1974*; *Whitfield and Evans, 1983*). Histological and ultrastructural studies defined these cells in miracidia and sporocysts by their stem cell-like morphology and rapid proliferation (*Schutte, 1974*; *Pan, 1980*). Recently, we showed that these germinal cells indeed drive proliferation within developing sporocysts and share some molecular signatures with stem cells from diverse organisms (*Wang et al., 2013*). In contrast, the cellular source of the schistosome germline, which underlies sexual reproduction in the mammalian host, remains an open question. Furthermore, because somatic stem cells were only recently identified in adult schistosomes (*Collins et al., 2013*), the relationships between germinal cells, germ cells, and somatic stem cells are unclear.

To clarify these relationships, we transcriptionally profiled stem cells from *Schistosoma mansoni* asexual (sporocyst) and sexual (juvenile) stages at both population and single-cell levels. We identified four transcriptionally distinct populations and validated this heterogeneity by in situ hybridization. By characterizing the behavior of these stem cells at major developmental transitions, we found that larvally derived stem cells serve as the source for the parasite's adult stem cells. We also identified a novel gene that is activated during development inside the mammalian host and serves as the

earliest marker for the schistosome germline. This work reveals the stem cell heterogeneity underlying the development and propagation of these important parasites.

## Results

### Single-cell RNAseq defines three major sporocyst stem cell classes

Each miracidium carries 10–20 germinal cells (*Pan, 1980*; *Cort et al., 1954*; *Wang et al., 2013*), which expand massively and differentiate to produce many daughter sporocysts (*Figure 1A*, and *Figure 1—figure supplement 1*). Our recent work has shown that germinal cells exhibit heterogeneity within this population (*Wang et al., 2013*), revealed by the distinct proliferation kinetics and expression of a schistosome homolog of *nanos* (*Wang and Lehmann, 1991*), a conserved regulator of germ cell development (*Juliano et al., 2010*; *Wang et al., 2007*) also expressed in the schistosome adult stem cells (*Collins et al., 2013*). To characterize this heterogeneity further, we isolated and transcriptionally profiled these stem cells from in vitro-transformed mother sporocysts (*Figure 1B*).

Principal component analysis (PCA) of single-cell transcriptomes revealed three major cell classes (*Figure 1C*). We designated these classes based upon their respective markers: κ-cells (kappa indicates $klf^+nanos-2^+$); φ-cells (phi indicates $fgfrA,B^+$); and δ-cells (delta indicates double-positive for *nanos-2* and *fgfrA,B*). The difference between κ and δ/φ-cells extends along PC1, and contributes to ~30% of the total variance among cells, whereas the difference between δ and φ-cells is secondary, delineated by PC2 and contributing ~10% of the total variance. For example, *nanos-2* exhibits almost equal loadings on both PCs, negative on PC1, positive on PC2, consistent with its expression in both κ and δ-cells. Based on projections along the first two PCs (*Treutlein et al., 2014*), we identified additional genes that contribute to the distinctions between classes: a schistosome *p53* homolog and a zinc finger protein (*zfp-1*) expressed abundantly in δ-cells and at lower levels in φ-cells; and a hes family transcription factor (*hesl*) expressed specifically in φ-cells (*Figure 1D and E*, *Supplementary file 1*). We validated these transcriptomic findings by fluorescent in situ hybridization (FISH) on in vitro-cultured mother sporocysts (*Figure 1—figure supplement 2*). Unfortunately, the κ class-specific marker *klf* was expressed at very low levels (*Figure 1E*), beneath the detection limits of our current FISH protocol.

In addition to these class-defining genes, the divergence of the three cell classes is manifested by hundreds of other genes that exhibit various levels of statistically significant differences between classes (*Figure 1—figure supplement 3*). However, these genes comprise only a small fraction of transcripts detected in these cells (N = 6,661), and most of them are not enriched in stem cells compared to differentiated cells. Notably, very few transcripts are specific to individual cell classes, with φ-cells showing the fewest specific markers. These observations confirm that sporocyst stem cells, regardless of the subpopulation to which they belong, share a common transcriptomic profile.

### Stem cell classes display distinct spatiotemporal patterns throughout asexual development

Examining *fgfrA* and *nanos-2* enabled us to distinguish all three cell classes in situ: φ-cells express *fgfrA*, κ-cells express *nanos-2*, and δ-cells express both. Thus, we followed these cells throughout intramolluscan development by monitoring *fgfrA* and *nanos-2* expression. After the first week of infection, asexually produced embryos– identified as compact, spherical cell clusters (*Schutte, 1974*) and from which daughter sporocysts will arise– begin to develop (*Figure 2A*). φ-cells were distributed beneath the parasite's outer layer and excluded from daughter embryos. δ-cells were found in large clusters within embryos. κ-cells clustered with δ-cells in embryos and were found in extraembryonic tissues as singlets or doublets, suggested to be the source of developing embryos in previous histological studies (*Schutte, 1974*).

Two weeks post-infection, mother sporocysts contain many mature daughter sporocysts that are ready to leave the mother and migrate elsewhere in the snail. At this stage κ-cells comprised the vast majority (>85%) of stem cells in these daughters (*Figure 2B*); fewer δ-cells were observed and φ-cells were mostly excluded. However, one week later, when post-migratory daughters colonized new regions of host tissue (*Figure 2C*), all three classes reappeared as intermingled populations, consistent with κ-cells generating the other stem cell types.

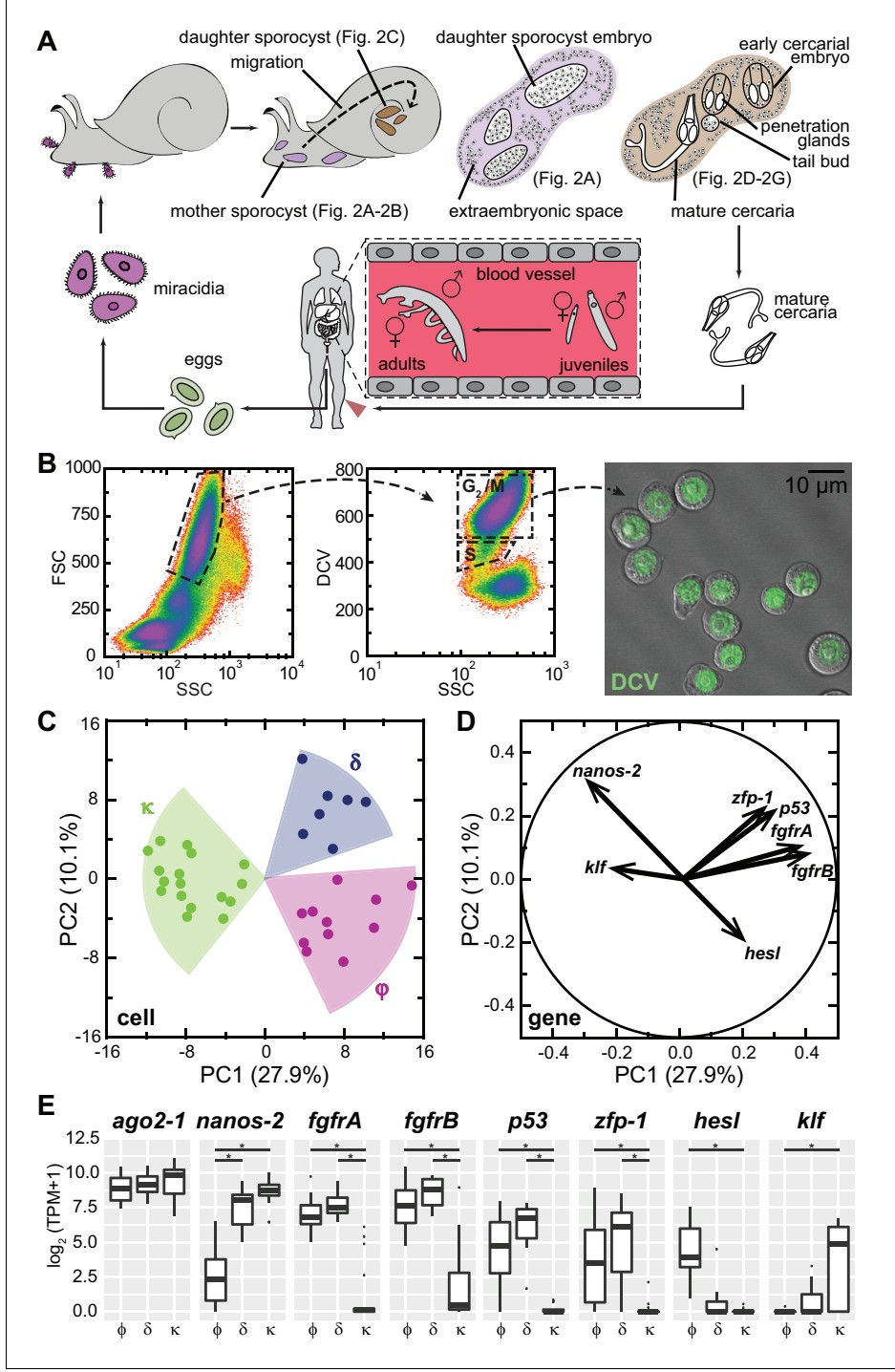

**Figure 1.** Single-cell RNAseq reveals stem cell classes in sporocysts. (**A**) Schematic of the schistosome life cycle. Images depicting developmental stages shown in *Figure 2* are labeled accordingly. (**B**) Dissociated cells were gated using forward scattering (FSC), side scattering (SSC), and DyeCycle Violet (DCV) fluorescence to isolate S or $G_2$/M phase cells from mother sporocysts. Dead cells and debris (<30% of total events) were pre-excluded based on high TOTO-3 fluorescence. Right: Sorted $G_2$/M phase cells from mother sporocysts visualized by DIC and fluorescence microscopy. (**C**) PCA of 35 single-cell transcriptomes of sporocyst stem cells. Summative variances are reported in percentages. Assignment of cell classes is based on hierarchical clustering. (**D**) Selected genes with heavy loadings are plotted in projection on the first 2 PCs. The projection on each axis represents the correlation coefficient of the respective gene with each principal component. (**E**) Box plots of expression levels of selected class-dependent genes. *ago2-1* expression is also shown as a ubiquitous stem cell marker. Boxes indicate quartiles

*Figure 1 continued on next page*

*Figure 1 continued*
and medians, whiskers show maxima and minima, and dots represent outliers (above and below 1.5X interquartile range). *p<0.01 (t-test). p-values were estimated based on multiple models using either TPM or log$_2$(TPM +1) as expression values.

DOI: https://doi.org/10.7554/eLife.35449.003

The following figure supplements are available for figure 1:

**Figure supplement 1.** *S. mansoni* sporocysts develop in *B. glabrata* snails.

DOI: https://doi.org/10.7554/eLife.35449.004

**Figure supplement 2.** Distribution of stem cell classes in in vitro-transformed mother sporocysts.

DOI: https://doi.org/10.7554/eLife.35449.005

**Figure supplement 3.** Number of genes differentially expressed between stem cell classes.

DOI: https://doi.org/10.7554/eLife.35449.006

Intramolluscan development culminates with the production of infectious cercariae. In early cercarial embryos (dashed circle in *Figure 2D–2E*), φ-cells were found concentrated both anteriorly and posteriorly, where the mouth and tail bud form, respectively. Additionally, two clusters of κ-cells were observed posterior to the penetration glands (*Figure 2D–2E*), at the site of the germinal cell cluster, considered gonadal primordia based on histological and ultrastructural studies (*Cheng and Bier, 1972*; *Dorsey et al., 2002*). In the mature cercarial body (*Figure 2F–2G*), the κ-cell pair posterior to the glands expands into two clusters that contain multiple cells each. In parallel, five δ-cells were detected in a regular pattern around the penetration glands, with one at the midline and two pairs laterally (*Figure 2E*), whereas φ-cells are absent at this stage. Since the cercarial body (but not the tail) penetrates the mammalian host, only δ and κ-cells, but not φ-cells, may be passed to the intramammalian (sexual) stage.

## Larvally derived stem cells drive initial proliferation in schistosomula

After emerging from the snail into water, cercariae burrow through mammalian host skin and their bodies transform into the next life-cycle stage, the schistosomula. At this stage, the parasites do not grow for several weeks, until they reach the hepatic portal vein. Thus, the extent of proliferation in the initial days after infection has been unclear (*Clegg, 1965*), with mitotic cells only detected 4 days post-infection (*Clegg and Smithers, 1972*). Furthermore, because the adult stem cells have only been identified recently (*Collins et al., 2013*), their developmental origin has yet to be investigated. The identification of δ and κ-cells in cercariae provides a potential source of new multipotent cells upon entry into the mammalian host.

We mimicked this transition by exposing cercariae to ex vivo mouse tail-skin biopsies and collecting transformed schistosomula on the other side of the skin (*Clegg and Smithers, 1972*; *Protasio et al., 2013*). Following skin penetration, we assayed proliferation in schistosomula via EdU labeling (*Figure 3A*). Between 22 and 36 hr post-transformation, we observed five EdU$^+$ cells around the penetration glands, anterior to the ventral sucker (*Figure 3B*), and confirmed that they were δ-cells (*fgfrA$^+$nanos-2$^+$*) (*Figure 3C*). During the next 12 hr, these cells completed mitosis, indicated by the appearance of five EdU$^+$ doublets (*Figure 3B*). Thereafter, the number of EdU$^+$ nuclei steadily increased (*Figure 3D*), but proliferation was restricted anteriorly to the ventral sucker, until one week later, when two clusters of ~2–3 EdU$^+$ cells appeared in the 'germinal cell cluster' region posterior to the ventral sucker, where κ-cells are found (*Figure 3B*, Days 9–10). EdU$^+$ cells were not detected in irradiated worms (*Figure 3E*), consistent with previous reports that irradiation leads to developmental defects and reduced pathogenicity (*Wilson, 2009*). These results suggest that a small, fixed number of κ and δ-cells are transmitted to the mammalian host. Because these cells appear to be the only dividing cells in schistosomula, they are likely the source of the recently identified stem cells in adult schistosomes (*Collins et al., 2013*; *2016*).

## Stem cells in juveniles reveal germline and somatic populations

Intramammalian growth initiates after schistosomula migrate into the portal vein, around 2 weeks post-infection (*Clegg, 1965*; *Basch, 1981*). To characterize proliferation driving juvenile growth, we harvested EdU-labeled parasites 3 weeks post-infection. Worms displayed a range of sizes based upon differences in arrival time, enabling a developmental time course to be reconstructed from a

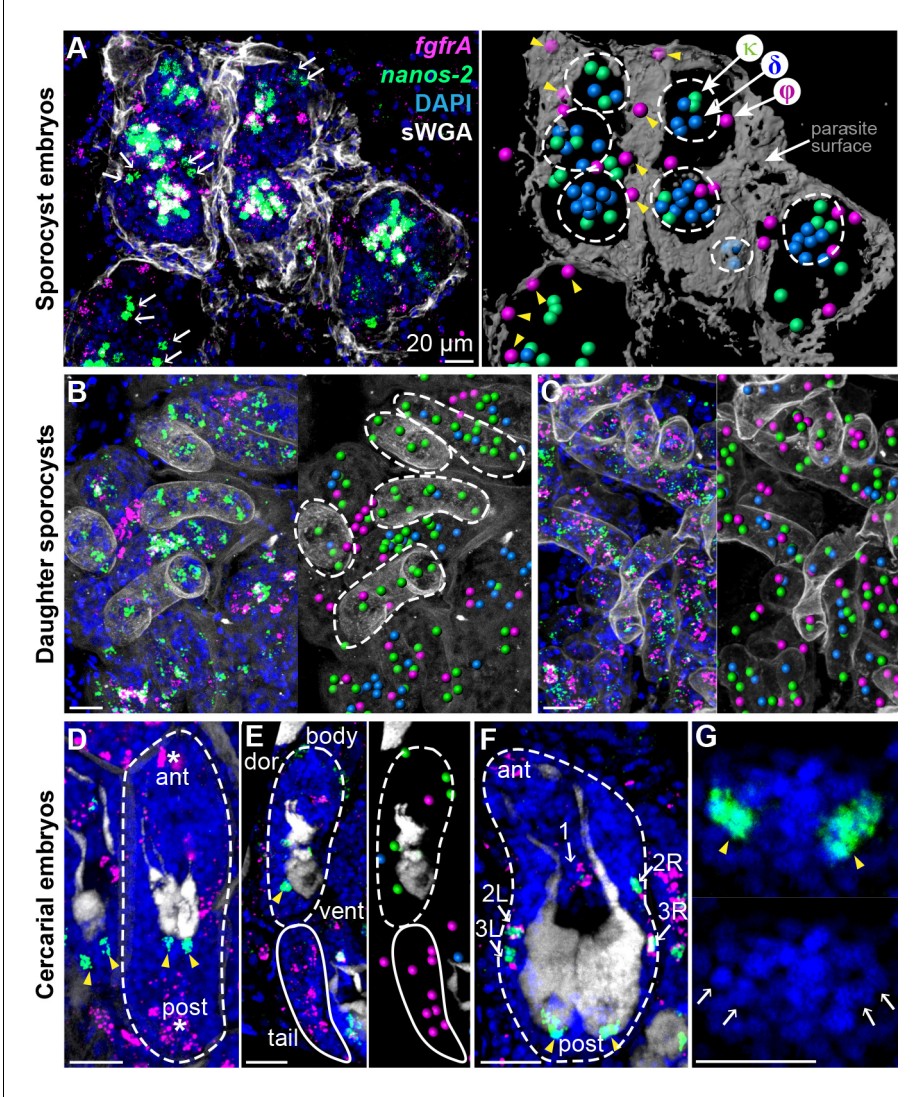

**Figure 2.** Stem cells exhibit class-specific spatial and temporal patterns during intramolluscan development. (A) FISH of *nanos-2* and *fgfrA* reveals spatial distributions of cell classes in a mother sporocyst containing daughter embryos (dashed circles), 10 days post-infection (dpi). sWGA (grey) labels the parasite surface (tegument). Arrows: extraembryonic κ-cells. Right: rendered image of that shown to the left. Spheres: cell centers; yellow arrowheads: φ-cells beneath the surface. (B) Mature daughter sporocysts contained in a mother sporocyst 15 dpi. Dashes outline the body surfaces of daughters. In daughters, κ-cells comprised 522 out of 592 counted stem cells in these daughters, while fewer δ-cells (61/592) and φ-cells (9/592) were observed. (C) Daughter sporocysts in snail ovotestis, 25 dpi. (D) Early cercarial embryo before the tail bud forms. φ-cells were found concentrated both anteriorly and posteriorly, where the mouth and tail bud form; only two κ-cells were detected posterior to the two penetration glands. (E) Later development during cercarial embryogenesis. φ-cells were mostly located in the tails; the κ-cell pair were posterior to the glands. (F) Mature cercarial body in daughter sporocysts 30 dpi (dorsal view). 1: anterior midline cell; 2L, 2R: anterior lateral cells; 3L, 3R: posterior lateral cells. Arrowheads: κ-cells, asterisks: φ-cells, arrows: δ-cells. sWGA: penetration glands. Note that only the cercarial body is transmitted to the mammalian host, whereas tails are discarded outside of the host epidermis during penetration. (G) κ-cell clusters, magnified from (F), contain multiple cells each (single confocal section). Arrows: individual nuclei in κ-cell clusters. Scale bars: 20 μm. All images are maximum intensity projections from 30 μm tissue cryosections. Since animals are thicker than the sections, parasite surface and penetration glands were used to determine the orientation and position of the sections.

DOI: https://doi.org/10.7554/eLife.35449.007

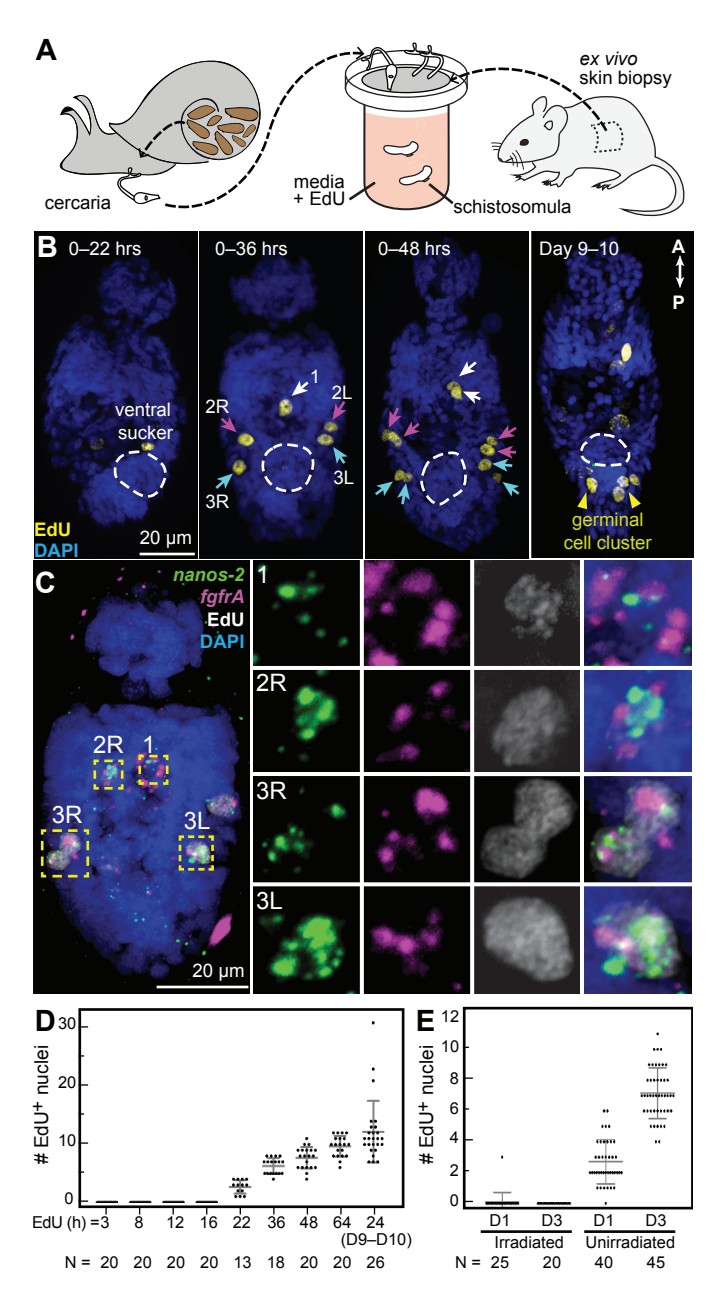

**Figure 3.** Larvally derived stem cells proliferate in schistosomula. (**A**) Schematic of in vitro transformation from cercariae to schistosomula and EdU labelling. (**B**) EdU⁺ cells are detected medially (1) and laterally (2L, 2R, 3L,3R) at the locations of δ-cells in cercarial bodies (ventral view). These cells divide to generate doublets, indicated by arrows. The time of EdU pulse post-transformation is indicated. Images are maximum intensity projections of confocal stacks. (**C**) Confocal maximum intensity projection of FISH of *nanos-2* and *fgfrA* on schistosomula at 2 days post-transformation confirms that only δ-cells incorporated EdU. Right: magnified images of boxed cells. (**D**) Quantification of EdU incorporation after transformation; x axis: length of EdU treatment post-transformation. Means and standard deviations are specified. N: number of worms analyzed. (**E**) Irradiated cercariae exhibit no EdU⁺ cells after transformation, which confirms that EdU specifically labels proliferating cells.

DOI: https://doi.org/10.7554/eLife.35449.008

static time point. EdU labeling revealed a posterior growth zone (PGZ) that extended as the parasites grew (*Figure 4A*). In more mature juveniles, all cells in primordial testes and ovaries also incorporated EdU (*Figure 4B*). To compare proliferating cells from juveniles to sporocyst stem cells, we transcriptionally profiled isolated cells undergoing division (S and $G_2$/M phase) from juveniles and sporocysts (*Figure 4—figure supplement 1*). Five-hundred and seventy-three genes were commonly enriched in both populations, including previously identified schistosome stem cell markers (e.g. *nanos-2*, *ago2-1*, and *fgfrA*), and cell cycle-associated transcripts (*h2a*, *cyclin B*, and *PCNA*) (*Wang et al., 2013*; *Collins et al., 2013*). Taken together, proliferating cells from juveniles resemble the sporocyst stem cells both morphologically and transcriptionally (*Figure 4—figure supplement 1*), suggesting that these cells represent juvenile stem cells, which support both somatic growth and germline development in the mammalian host.

Are κ and δ-cells maintained during juvenile development? Since κ and δ-cells can be distinguished by a small set of transcripts, we measured expression of 87 genes, including all identified

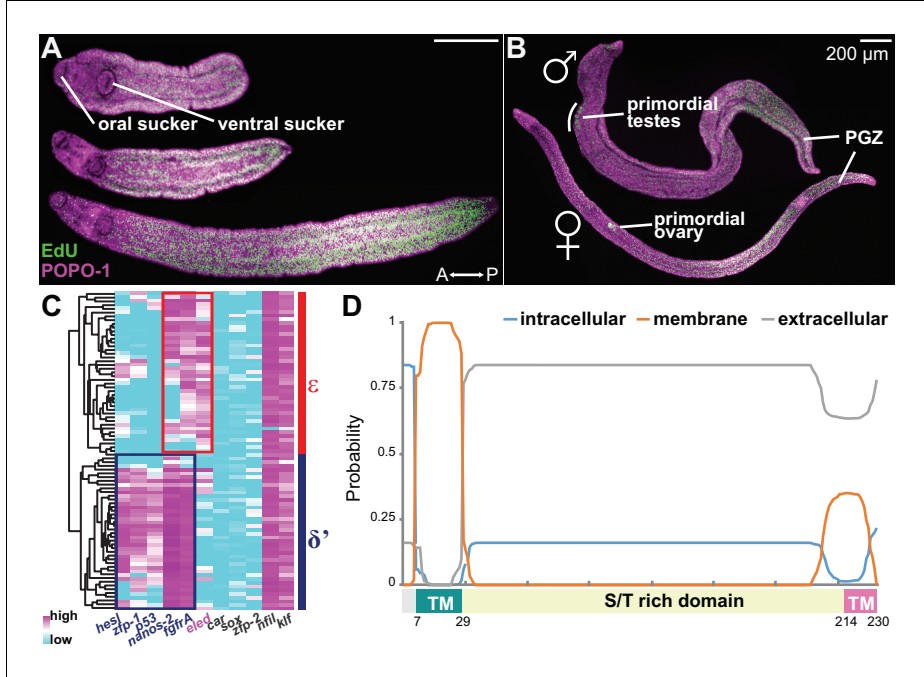

**Figure 4.** Emergence of a new stem cell class, ε, in juveniles. (A–B) Distributions of EdU+ cells in three-week-old juveniles. Note the high density of EdU+ cells toward the posterior of the worms. (B) More mature males with partly developed lateral body extensions and females with a visible uterus display EdU+ cells in primordial gonads and the posterior growth zone (PGZ). (C) Hierarchical clustering of 85 juvenile stem cells distinguishes two major cell classes. Gene names in blue: cell class-dependent genes identified in the sporocyst stem cells; gene names in grey: top genes upregulated in juvenile stem cells compared to sporocyst stem cells. Expression levels were standardized gene-by-gene by mean-centering and dividing by the standard deviation of expressing cells. (D) Domain diagram of Eled, predicted by TMHMM 2.0 (*Krogh et al., 2001*). TM: transmembrane domain. S/T rich domain: extracellular domain enriched in serine/threonine.

DOI: https://doi.org/10.7554/eLife.35449.009

The following figure supplements are available for figure 4:

**Figure supplement 1.** RNAseq comparison of FACS-isolated proliferating cells from sporocysts and juveniles.
DOI: https://doi.org/10.7554/eLife.35449.010

**Figure supplement 2.** Analysis of technical variation of single-cell qPCR.
DOI: https://doi.org/10.7554/eLife.35449.011

**Figure supplement 3.** Single-cell qPCR identifies ε-cells in juveniles.
DOI: https://doi.org/10.7554/eLife.35449.012

**Figure supplement 4.** *eled* gene structure.
DOI: https://doi.org/10.7554/eLife.35449.013

cell class-specific factors, across single juvenile proliferating cells by multiplex qPCR (*Figure 4—figure supplement 2*, and *Supplementary file 2*) (*van Wolfswinkel et al., 2014*). The assayed gene set contained the class-specific factors identified from sporocyst stem cells and the most highly enriched genes in juvenile vs. sporocyst stem cells. Hierarchical clustering identified two major cell classes (*Figure 4C*). δ'-cells are similar to δ-cells, but express abundantly both δ and φ-cell markers (including *nanos-2*, *fgfrA*, *p53*, *zfp-1*, and *hesl*), indicating that these cells are the likely source of the adult somatic stem cells (*Collins et al., 2013*; *2016*).

The other class abundantly expresses a novel schistosome-specific factor, *eledh (eled)*, which is undetectable in sporocysts but is among the most abundant transcripts in juvenile stem cells (*Supplementary file 3*). We designated this class ε (epsilon indicates *eled*[+]), which displays lower expression of *nanos-2* and *fgfrA*, and similar to κ-cells, lacks expression of *p53*, *zfp-1*, and *hesl* (*Figure 4C*, and *Figure 4—figure supplement 3*). Based on these similarities in gene expression, we propose that ε-cells are likely derivatives of κ-cells (see Discussion for more details). *eled* is a single-copy gene (Smp_041540) that was previously annotated as a nuclear hormone receptor, *dhr4* (*Protasio et al., 2012*; *Berriman et al., 2009*); however, our analysis suggests that this gene does not encode a hormone receptor (*Figure 4—figure supplement 4*). As described below, this gene antagonizes *nanos* (Greek for 'dwarf'); therefore, we named Smp_041540 *eledh* (Sindarin for 'elf'), based on the antagonistic relationship between dwarves and elves in Tolkien's world. Homologs of *eled* are found across *Schistosoma* species, whereas planarian and tapeworm genomes appear to lack them. *Figure 4D* shows the predicted architecture of *eled*, which has a transmembrane domain at the N-terminus (probability ~100%), a putative transmembrane domain at the C-terminus (probability ~35%), and an extracellular domain in between (probability ~85%). The extracellular domain contains 19% serine and 17% threonine, presenting the highest serine/threonine fraction in known proteins.

In juveniles *eled* expression was detected in primordial testes, ovaries, and vitellaria, as well as in a gradient increasing toward the PGZ, which lacks reproductive organs in males (*Figure 5A*). This pattern is distinct from that of the proliferation marker, *h2a*, which labels all stem cells and is more evenly distributed. Quantitatively, overlay of *eled* and *h2a* revealed that all gonadal *eled*[+] cells (*Figure 5B*) and ~80% of somatic *eled*[+] cells (966/1269, from two male juveniles) were also *h2a*[+]. Conversely, all germ cells within gonadal primordia were *eled*[+], but only ~50% of the somatic *h2a*[+] cells in the PGZ (1679/3576) and ~10% of the somatic *h2a*[+] cells outside the PGZ (769/5366) expressed *eled*. In mature adults, *eled* expression was limited to reproductive organs (*Figure 5C*). Thus, somatic ε-cells may be a transitory source of new tissue underlying massive juvenile growth in the PGZ.

## *eled* is the earliest schistosome germline marker and functions in opposition to *nanos*

Expression of *eled* in gonadal primordia led us to examine earlier stages of germ cell development. We found that *eled* expression precedes *nanos-1*, which is germline specific both in juveniles (*Figure 6—figure supplement 1*) and adults (*Iyer et al., 2016*; *Wang and Collins, 2016*): only a subset of *eled*[+] cells in gonadal primordia co-express *nanos-1*, and the number of *eled*[+]*nanos-1*[+] cells increases over the course of development (*Figure 6A,B*). Quantification reveals a sharp transition as worm length exceeds ~400 μm: before the transition, none of the gonadal *eled*[+] cells is *nanos-1*[+]; after the transition, most if not all of the gonadal *eled*[+] cells become *nanos-1*[+] (*Figure 6C*). These results suggest that germ cells may be derived from ε-cells early in juvenile development, and *eled* is the earliest germline marker yet identified in schistosomes.

To characterize functional interactions between *eled* and *nanos*, we knocked down gene function using RNA interference (RNAi) (*Mann et al., 2010*; *Collins et al., 2013*; *Wang et al., 2013*). For these experiments, juveniles were soaked in double-stranded RNA (dsRNA) continuously in vitro for 2 weeks. We focused on male juveniles because female development was retarded under in vitro culture. To assess gene expression changes after RNAi, we performed whole-mount in situ hybridization (WISH); we used confocal microscopy to examine testis structure.

*eled* RNAi resulted in upregulation of *nanos-2* in the PGZ (*Figure 7A*), where the somatic ε- cells are located. Knockdown of *nanos-1* or *nanos-2* resulted in degenerated testes and loss of differentiated germ cells; however, the remaining gonads maintained *eled* expression (*Figure 7A and B*). By contrast, knockdown of *eled* resulted in premature accumulation of sperm in juvenile testes

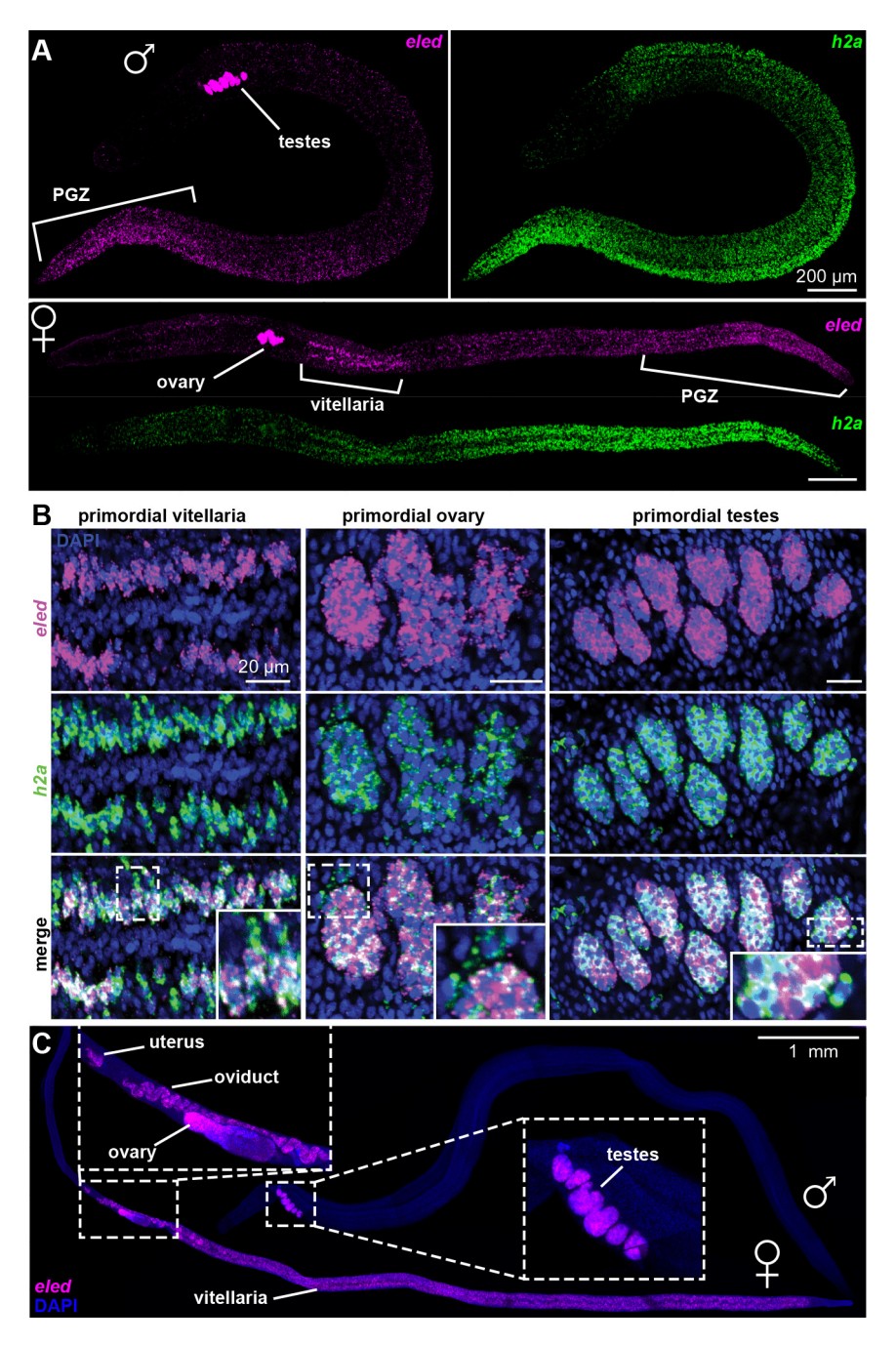

**Figure 5.** Changes in *eled* expression during the course of intramammalian development. (**A**) FISH detects *eled* and *h2a* expression in juveniles. The boundary of the PGZ is defined by a sharp drop of *eled*[+] cells. *eled* signal in testes and ovary was over saturated in order to detect weaker expression in soma. (**B**) Double FISH of *eled* and *h2a* in juvenile gonads. Insets: magnified boxed areas. (**C**) *eled* expression in adults. *eled*[+] cells were only detected in reproductive organs (insets). Note high expression in testes and anterior ovary, where oogonial stem cells are found, but low expression in the posterior ovary, where germ cells differentiate.
DOI: https://doi.org/10.7554/eLife.35449.014

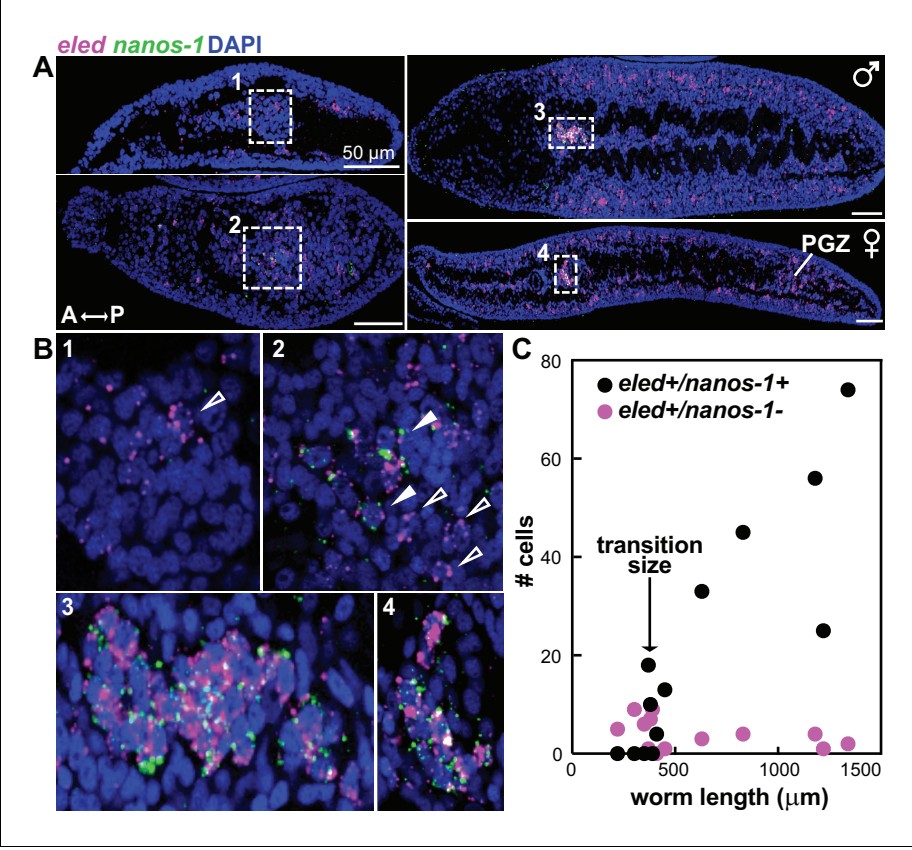

**Figure 6.** *eled* expression precedes that of germline-specific *nanos-1*. (**A**) Double FISH of *eled* and *nanos-1* in 3-week juveniles, showing the emergence of *nanos-1* expression in gonadal primordia. (**B**) Magnified boxed gonadal regions in (**A**). Images are numbered from less to more mature worms. Images are confocal sections. Empty arrowheads: $eled^{+}nanos\text{-}1^{-}$; solid arrowheads: $eled^{+}nanos\text{-}1^{+}$. (**C**) Quantification of *nanos-1* expression in $eled^{+}$ presumptive germ cells as a function of worm length. Symbols represent cell counts in individual animals. Note the sharp transition at worm length ~400 µm (N = 13).

DOI: https://doi.org/10.7554/eLife.35449.015

The following figure supplement is available for figure 6:

**Figure supplement 1.** *nanos-1* expression is germline specific.

DOI: https://doi.org/10.7554/eLife.35449.016

(*Figure 7B*). Thus, *eled* appears to inhibit, whereas *nanos* genes are required for, germ cell differentiation. Together, these results suggest that *eled* antagonizes the two schistosome *nanos* homologs: in the soma, it suppresses *nanos-2* expression in ε-cells in the PGZ; in the germline, it inhibits germ cell differentiation.

## Discussion

Throughout their life cycles, parasitic flatworms undergo dramatic morphological changes as they switch from free-living, infectious stages to endoparasitic forms residing in different hosts; the cellular basis of this developmental plasticity is largely unknown. Here, we have used single-cell transcriptional profiling to characterize schistosome stem cells during intramolluscan and intramammalian development. We found that these stem cells are a heterogeneous population, consisting of four major classes, distinguished by several distinct markers. This analysis enabled cellular and molecular studies to trace the origin of the adult stem cells back to a handful of larvally derived cells packaged into the infectious stage; it also provided evidence for the origin of the schistosome germline.

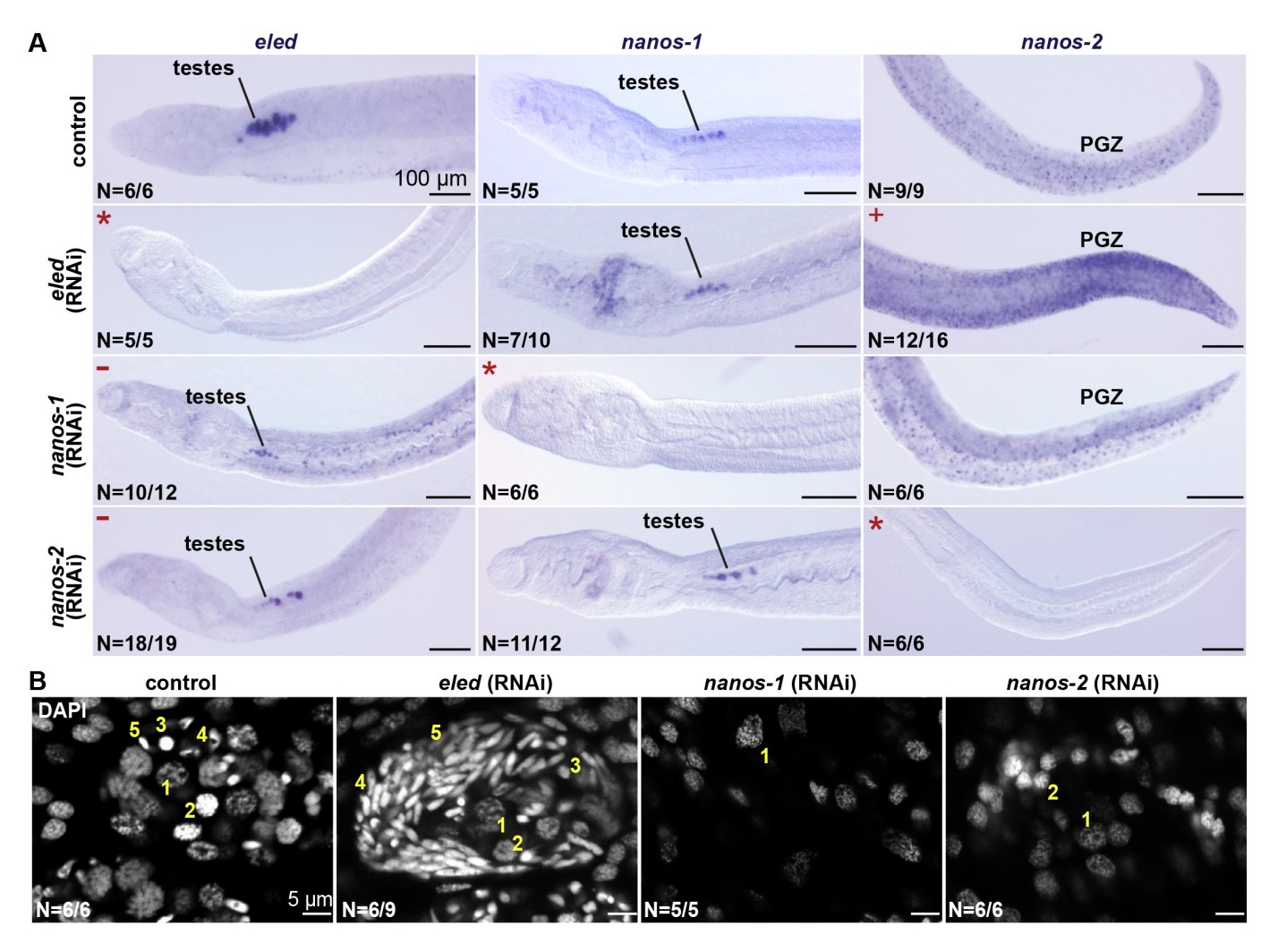

**Figure 7.** *eled* functionally antagonizes *nanos*. (**A**) WISH images showing the expression of *eled*, *nanos-1*, and *nanos-2* in male juveniles after RNAi. +: higher *nanos-2* expression compared to the control; -: regressed testes; asterisks: RNAi depletion of the target transcripts. (**B**) Confocal sections of DAPI-stained testes in RNAi worms. Nuclear morphologies are consistent with (1) undifferentiated spermatogonium, (2) spermatocyte; (3) round spermatid, (4) elongating spermatid, and (5) sperm. In *nanos-1* or *nanos-2* (RNAi) animals, all testis lobes regressed; whereas in *eled* (RNAi) animals, worms with more than two testis lobes with accumulation of sperm were counted as manifesting the phenotype (schistosomes normally possess 6–8 testis lobes). N: number of worms analyzed in one biological replicate; penetrance was consistent between 4 biological replicates.
DOI: https://doi.org/10.7554/eLife.35449.017

During intramolluscan asexual development we find three classes of stem cells that can be distinguished by expression of either or both *nanos-2* and *fgfr*: κ (*klf*+*nanos-2*+); φ (*fgfrA*+); and δ (*nanos-2*+*fgfrA*+) cells. Our data are consistent with the proposed lineage depicted in *Figure 8*. We posit that κ cells serve as 'embryonic' stem cells: in addition to *nanos-2* they express a *klf* homolog and are found isolated in developing mother sporocysts. This observation is consistent with classic histological studies suggesting that such isolated germinal cells serve as the source of developing embryos (*Schutte, 1974*; *Pan, 1965*). These studies reported that the scattered stem cells may persist in mother sporocysts for months and continuously undergo asymmetric divisions, with one daughter cell forming an embryo while the other remains undifferentiated (*Schutte, 1974*). We found that κ-cells are dramatically enriched in mature, pre-migratory daughter sporocysts, consistent with the idea that these cells generate the other stem cell types observed in post-migratory daughter sporocysts.

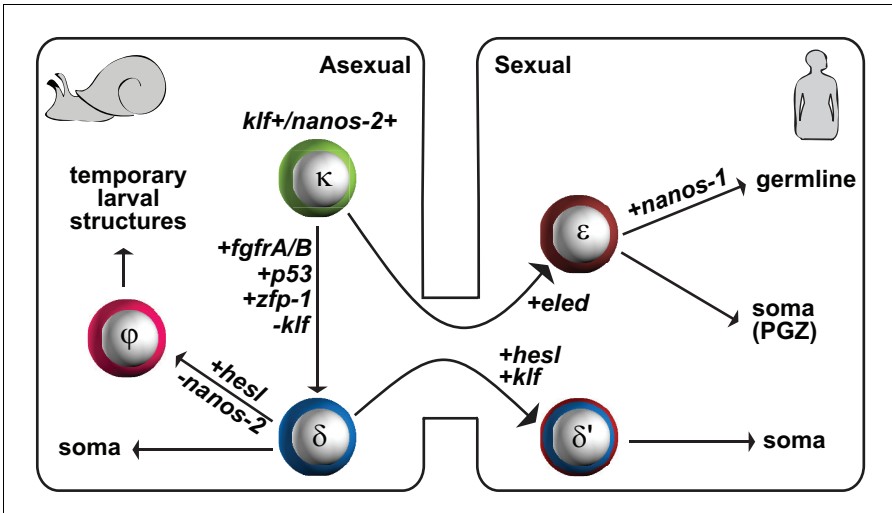

**Figure 8.** A proposed model for the schistosome stem cell classes. κ-cells are at the top of the hierarchy, and express *nanos-2* and *klf*. Activation of several genes associated with somatic stem cell function in adults (*fgfrA/B; p53; zfp-1*) leads to the specification of δ-cells, which we propose serve to generate somatic tissues. Downregulation of *nanos-2* and activation of *hesl* in δ-cells leads to the formation of φ-cells, which are associated with many transitory larval structures, including the sporocyst epidermis (tegument) and the cercarial tail. Only a small number of κ and δ-cells is transmitted from the asexual to the sexual life-cycle stages. After entry into the mammalian host, activation of *hesl* in δ-cells leads to δ'-cells, which appear to serve as the source of the adult somatic stem cells. κ-cells downregulate *nanos-2* and activate expression of an intramammalian stage-specific transcript, *eled*, generating ε-cells, which are distributed in gonadal primordia and the posterior growth zone. During maturation, gonadal ε-cells subsequently activate germline-specific *nanos-1*, which commits ε-cells to germline fate.

DOI: https://doi.org/10.7554/eLife.35449.018

Activation of several genes associated with somatic stem cell function in adults (e.g. *fgfrA/B; p53; zfp-1*) leads to the specification of δ-cells from κ-cells; we propose that δ-cells serve to generate somatic tissues. We suggest that downregulation of *nanos-2* and activation of *hesl* in δ-cells leads to the formation of φ-cells. The restricted distribution of φ-cells in transitory larval structures, including the sporocyst epidermis (tegument) and the cercarial tail (*Figure 2A,E*), is consistent with the observations that φ-cells are not transmitted between life-cycle stages and are absent during the sexual stage. Our previous work identified rapidly cycling stem cells that do not express *nanos-2* (in contrast to slower-cycling cells that do) (*Wang et al., 2013*); thus, φ-cells appear to be a lineage-committed, transit-amplifying population that produces temporary larval tissues. The distinction between κ, δ, and φ-cells is also consistent with early histological studies that observed germinal cells in three characteristic tissue locations within mother sporocysts: either scattered among daughter embryos, clustered inside of embryos, or situated close to mother sporocyst walls (*Schutte, 1974*; *Pan, 1965*).

The result of larval development in the snail is the production of cercariae, free-swimming infectious forms that penetrate mammalian skin to continue the life cycle. We observed five δ-cells, localized in a stereotypical pattern in the cercarial body. Using mouse skin explants to mimic the transformation that occurs after parasites enter their mammalian host, we found that these five cells proliferate during the first 24 hr after transformation. Given the similarities between the transcriptional profiles of δ-cells and the somatic stem cells found in juvenile parasites (δ'-cells that also express *hesl*), as well as their early proliferation upon penetrating host skin, we suggest that δ-cells serve as the source of the somatic stem cells identified in adult schistosomes. Because there is no growth during the first 2 weeks post-infection (*Clegg, 1965*; *Wilson, 2009*), these early-proliferating cells may contribute to the tissue remodeling (e.g. of the tegument or digestive system) required for the parasite's subsequent growth and maturation.

Cercariae also contain two clusters of κ-cells, in the germinal cell cluster that is proposed to be the gonadal primordia (*Schutte, 1974*; *Dorsey et al., 2002*). These cells do not proliferate in the initial days after transformation; instead, we detect proliferation in the posterior of the schistosomula 9–10 days after transformation in vitro. Based on the antagonistic interaction between *eled* and *nanos*, we suggest that κ-cells downregulate *nanos-2* and activate expression of *eled*. *eled* is specific to the intramammalian stage of the life cycle and defines ε-cells. The absence of δ/δ'-cell markers, *fgfr*, *zfp-1*, *p53*, and *hesl*, in both κ and ε-cells favors the model that κ/ε-cells form a separate lineage from δ/δ'-cells. Expression of *eled* is dynamic: during juvenile development, *eled* is expressed in gonadal primordia and in a subset of proliferating cells in the posterior growth zone. In mature adults, *eled* expression is maintained in the reproductive system but is no longer detected somatically. Thus, ε-cells can either give rise to germline or to a transient somatic population that appears to drive the massive posterior growth exhibited by these parasites. During maturation, gonadal ε-cells subsequently activate germline-specific *nanos-1*. We suggest that the activation of *nanos-1* commits ε-cells to germline fate. How *eled*$^+$ cells choose to generate germline (*nanos-1*$^+$) vs. posterior somatic cells (*nanos-1*$^-$) will be an important avenue for future studies.

Functional analysis of *eled* was limited to juvenile male parasites harvested after 3 weeks of growth in the mouse, due to technical constraints imposed by the parasite's life cycle. Under these experimental conditions, *eled* (RNAi) males exhibited premature production of sperm; thus, *eled* appears to act as a brake on germ cell differentiation. Because it is also the earliest germline marker yet identified and could play a role in germ cell specification, new experimental techniques will have to be developed to analyze gene function during the first 3 weeks of development in the mammalian host, when the germline is formed. Similarly, rigorous experimental testing of the proposed lineage relationships between the stem cell classes (*Figure 8*) will require introduction of lineage-tracing techniques in these parasites.

Previous work noted similarities between schistosome stem cells and the neoblasts that drive regeneration in free-living planarians (*Wang et al., 2013*; *Collins et al., 2013*). The heterogeneity we detected in schistosome stem cells is also reminiscent of that observed in the planarian neoblasts (*van Wolfswinkel et al., 2014*), and we observed a striking overlap in a group of genes co-regulated between stem cell classes from both organisms. The genes *fgfr*, *zfp-1*, and *p53* were defined as markers of an epidermally committed population in planarians (*van Wolfswinkel et al., 2014*). These genes are abundantly expressed in δ/δ'-cells, the major somatic population in schistosomes. Recently, schistosome adult stem cells were shown to have a strong differentiation bias toward the tegumental lineage (*Collins et al., 2016*); the necessity of *zfp-1* family genes for proper tegumental fate or, more generally, for proper differentiation, further linked these parasites to their free-living ancestors (*Wendt et al., 2018*). Furthermore, in the planarian embryo, blastomeres produce temporary embryonic tissues but also give rise to neoblasts by downregulating a set of embryo-specific genes and upregulating genes associated with adult development (e.g., *zfp-1* and *p53*) (*Davies et al., 2017*). Our data suggest that a similar transition may also occur between κ-cells and δ/δ'-cells. Beyond planarians and schistosomes, stem cells have also been described in other parasitic flatworms, including tapeworms (*Hoffmann et al., 2014*; *Koziol and Brehm, 2015*; *McCusker et al., 2016*). We expect that single-cell approaches applied to additional parasitic flatworms will provide a broader overview of the role of stem cell heterogeneity in driving such complex life cycles.

This study presents an important step toward understanding the fundamental mechanisms driving the propagation and long-term survival of schistosomes. As the causative agents of a neglected tropical disease impacting hundreds of millions of people, these parasites present a major threat to global health. Schistosome infection is currently treated with a single drug, praziquantel, which is used in mass drug administration programs (*Hoffmann et al., 2014*; *Sokolow et al., 2015*). With concerns about resistant strains emerging (*Hoffmann et al., 2014*), it becomes increasingly important to understand the fundamental mechanisms driving the propagation and long-term survival of these parasites. Characterizing the roles of the stem cell populations defined here in schistosome transmission, reproductive development, and survival may ultimately lead to novel approaches to reduce the burden imposed by these parasites (*Matthews, 2011*; *Valentim et al., 2013*; *Hoffmann et al., 2014*).

# Materials and methods

## Key resources table

| Reagent type (species) or resource | Designation | Source or reference | Identifiers | Additional information |
|---|---|---|---|---|
| Strain (*Schistosoma mansoni*) | NMRI | BEI Resources | NR-21963 | |
| Antibody | Anti-Digoxigenin-POD | Roche | 11207733910 | |
| Antibody | Anti-Digoxigenin-AP | Roche | 11093274910 | |
| Antibody | Anti-DNP-HRP | PerkinElmer | FP1128 | |
| Antibody | Anti-Fluorescein-POD | Roche | 11426346910 | |
| Recombinant DNA reagent | Plasmid-pJC53.2 | Addgene | 26536 | |
| Chemical compound | succinylated Wheat Germ Agglutinin (sWGA) | Vector Laboratories | FL-1021S | |
| Chemical compound | Carboxyrhodamine 110 azide | Click Chemistry Tools | AZ105 | |
| Chemical compound | Alexa Fluor 488 azide | Invitrogen | A10266 | |
| Chemical compound | 5-ethynyl-2-deoxyuridine (EdU) | Invitrogen | A10044 | |
| Chemical compound | Vybrant DyeCycle Violet (DCV) | Invitrogen | V35003 | |
| Chemical compound | POPO-1 | Invitrogen | P3580 | |
| Chemical compound | TOTO-3 | Invitrogen | T3604 | |
| Chemical compound | Calcein AM | Invitrogen | C3100MP | |

## Parasite harvesting

In vitro-transformed mother sporocysts were obtained as detailed previously (*Wang et al., 2013*; *Mann et al., 2010*; *Ivanchenko et al., 1999*). Briefly, *S. mansoni* (strain NMRI) eggs were purified from livers harvested from schistosome-infected mice (Swiss Webster, female, ~7 weeks post-infection). Free-swimming miracidia were hatched from eggs in artificial pond water and transformed in vitro to mother sporocysts by exchanging pond water with sporocyst culture medium supplemented with 1X Antibiotic-Antimycotic (Gibco) and 20 µg/mL gentamycin (Gemini) at 37°C in 5% $CO_2$/5% $O_2$ for 48 hr. *S. mansoni* cercariae were shed from infected *Biomphalaria glabrata* snails about 5–8 weeks post-infection by exposing snails to bright light at 26°C for 1–2 hr. Schistosomula were transformed from cercariae using skin transformation (*Clegg and Smithers, 1972*; *Protasio et al., 2013*), in which cercariae were placed on ex vivo mouse skin biopsies (Swiss-Webster, Taconic) overlaid on Basch medium 169 (*Basch, 1981*) to collect parasites that passed through the skin. Juvenile and adult worms were obtained from infected mice (Swiss Webster NR-21963, ~3 or 6–7 weeks post-infection, respectively) by hepatic portal vein perfusion using 37°C DMEM. Worms were cultured at 37°C/5% $CO_2$ in Basch Medium 169 supplemented with 1X Antibiotic-Antimycotic.

In adherence to the Animal Welfare Act and the Public Health Service Policy on Humane Care and Use of Laboratory Animals, all experiments with and care of mice were performed in accordance with protocols approved by the Institutional Animal Care and Use Committees (IACUC) of: Stanford University (protocol approval number 30366); University of Illinois at Urbana-Champaign (protocol approval number 15134); University of Wisconsin–Madison (protocol approval number M005569).

## Culture media

Artificial pond water: 0.125 mg/L $FeCl_3 \cdot 6H_2O$, 32.25 mg/L $CaCl_2 \cdot 2H_2O$, 25 mg/L $MgSO_4 \cdot 7H_2O$, 42.5 mg/L $KH_2PO_4$, 1.875 mg/L $(NH_4)_2SO_4$, pH 7.2.

Medium F (*Mann et al., 2010*; *Ivanchenko et al., 1999*): 1X BME vitamins, 1X BME amino acids, 6 mg/L serine, 2.9 mg/L proline, 2.4 mg/L L-alanine, 2.8 mg/L aspartic acid, 4.7 mg/L glutamic acid, 2.4 mg/L glycine, 2.4 mg/L β-alanine, 40 mg/L malic acid, 30 mg/L ketoglutaric acid, 10 mg/L succinic acid, 5 mg/L fumaric acid, 10 mg/L citric acid, 70 mg/L $Na_2HPO_4$, 0.53 g/L $CaCl_2 \cdot 2H_2O$, 0.15 g/L KCl, 0.45 g/L $MgSO_4 \cdot 7H_2O$, 1.5 g/L NaCl, 4.5 g/L galactose, 1 g/L glucose, 25 mM HEPES, pH 7.

Sporocyst culture medium is modified from *Ivanchenko et al. (1999)*: 10% heat inactivated FBS, 23.5% Medium F, 23.5% DMEM/F12, 10% Schneider's Drosophila Medium, 2 g/L lactalbumin hydrolysate, 0.6 g/L galactose, 55 μM 2-mercaptoethanol, 0.005% Chemically defined lipid (Invitrogen).

Modified Basch 169 medium (*Mann et al., 2010*): 1 g/L lactalbumin hydrolysate, 1 g/L glucose, 8 mg/L insulin, 1 μM serotonin, 1 μM hydrocortisone, 0.5 μM hypoxanthine, 0.2 μM triiodothyronine, 0.5X MEM vitamins, 5% Schneider's Drosophila Medium, 10% heat inactivated FBS, 10 mM HEPES, pH 7.

## Parasite treatments

To ablate dividing cells, cercariae or juveniles were exposed to either 200 Gy of γ-irradiation on a Gammacell-220 Excel with a $Co^{60}$ source (Nordion) or 250 Gy of X-ray irradiation on a CellRad Faxitron source and cultured for 48 hr for juveniles and 72 hr for schistosomula.

For EdU labeling, schistosomula were pulsed with 2–5 μM EdU. Juveniles were pulsed with 10 μM EdU overnight. EdU incorporation was detected by click reaction with 20–50 μM Alexa Fluor azide conjugates (Invitrogen) as described previously (*Wang et al., 2013*; *Collins et al., 2013*). Experiments were confirmed on three biological replicates, each containing a cohort of juvenile worms (N =~50) collected from a separate infection. For schistosomula, two to three independent replicates were performed, each batch with ~10–15 worms for each condition. Confirming the specificity of EdU labeling, $EdU^+$ cells were not detected in irradiated schistosomula or juveniles (*Figure 3D*, and *Figure 4—figure supplement 1C*). For infected snails, animals were fed with 'gel food' containing 60 mg/mL spiruline and 1 mg/mL low-melting agarose in artificial pond water supplemented with 40 μM EdU (Invitrogen). Following the EdU pulse, the snails were washed, cultured at 26°C for 1–2 d, and fixed as described in the previous section.

RNAi was performed using previously described protocols (*Wang et al., 2013*; *Collins et al., 2013*). Clones were generated using oligonucleotide primers listed in *Supplementary file 3*. For RNAi, juveniles were soaked in ~20 μg/mL dsRNA for 2 weeks, with media containing dsRNA refreshed daily. We noticed that worm density is critical to achieve efficient knockdown. We used 20–30 juveniles per mL of medium. Each RNAi was repeated on at least three biological replicates (each replicate is from a separate infection). Every biological replicate contained two technical replicates (each replicate is one well of a 24-well plate and contains 20–30 worms). In rare situations, wells with juveniles showing significantly lower activities at the end of RNAi treatments were excluded from downstream analysis, as this deterioration in overall physiology is likely caused by poor culture conditions. To assess gene expression changes after RNAi, we performed WISH as signal development can be quenched while still in the linear range. The development was performed in parallel in control and RNAi animals and stopped simultaneously. Imaging was performed with identical illumination and exposure settings.

## In situ hybridization

RNA FISH experiments were performed as detailed previously (*Wang et al., 2013*; *Collins et al., 2013*) with modifications specific to each life-cycle stage. To observe various intramolluscan stages, schistosome-infected *B. glabrata* snails (10, 15, 25, or 30 days post-infection, dpi) were relaxed in sodium pentobarbital solution (0.5 mg/mL) for 6 hr, killed in hot water (70–90°C) for 30 s, deshelled, and fixed in 4% formaldehyde in artificial pond water supplemented with 0.2% Triton X-100% and 1% NP-40 for 24 hr at 4°C. The snail tissue was then bleached in a formamide bleaching solution (0.5% formamide, 0.5% SSC, and 1.2% $H_2O_2$) for 90 min, equilibrated in 30% sucrose/PBSTx (PBS with 0.3% Triton X-100) overnight, embedded in TBS tissue freezing medium, and cryosectioned at 30 μm thickness. Dried cryosections of snail tissues were then rehydrated in PBSTx on gelatin-coated slides, permeabilized by 2 μg/mL proteinase K for 5 min, and post-fixed for 10 min. Schistosomula were killed with ice-cold 1% HCl for 30–60 s before fixation. Fixed, dehydrated in vitro-transformed mother sporocysts were rehydrated, permeabilized by 2 μg/mL proteinase K (proK, Invitrogen) for 5 min, and post-fixed for 10 min in 4% formaldehyde in PBSTx. Schistosomula were bleached in the formamide bleaching solution for 10 min, and permeabilized by 5 μg/mL proteinase K for 10 min. Juveniles were killed in 6 M $MgCl_2$ for 30 s-5 min, fixed for 4 hr, dehydrated in methanol, incubated in 3% $H_2O_2$ in methanol for 30 min, then rehydrated, permeabilized by 10 μg/mL proteinase K for 10 min, and post-fixed. Adults were permeabilized by 5 μg/mL proteinase K for 45 min.

The hybridization step was carried out at 52°C overnight as described previously (*Wang et al., 2013*; *Collins et al., 2013*). Following washes, samples were blocked with 5% heat-inactivated horse serum and 0.5% Roche Western Blocking Reagent in PBSTx, and then incubated with antibody peroxidase conjugates at 4°C overnight. Detection was performed using tyramide signal amplification (TSA) with lab-made reagents (*King and Newmark, 2013*). For double FISH, the first peroxidase reaction was quenched for 30 min in 0.1% sodium azide solution before detection of the second gene. Imaging was performed in *scale* solution (30% glycerol, 0.1% Triton X-100, 4 M urea in PBS supplemented with 2 mg/mL sodium ascorbate). Clones used for riboprobe and dsRNA synthesis were generated as described previously (*Wang et al., 2013*; *Collins et al., 2013*), with oligonucleotide primers listed in *Supplementary file 3*. WISH follows the same procedure of FISH, except that the detection was carried out using antibody phosphatase conjugates for chromogenic development with NBT/BCIP (Sigma).

All FISH/WISH experiments were repeated on at least three biological replicates, each from a separate infection. All expression patterns throughout developmental stages were confirmed on multiple animals, specifically, ~150 in vitro-transformed mother sporocysts, ~5 mother and daughter sporocysts in snails, ~20 cercarial embryos, ~10 schistosomula, ~10 juveniles, and ~5 adults per biological replicate. For intramolluscan stages, as parasites are thicker than 30 μm (the cryosection thickness), we used parasite surface and penetration glands to determine the orientation and the position of the sections.

To assign cell classes using FISH signals, confocal stacks were obtained from a laser-scanning microscope using over-sampled resolutions recommended by Imaris (Bitplane). The stacks were resampled to give isotropic voxels, and subjected to Gaussian filtering and background subtraction. Center of labeled cell bodies was segmented channel-by-channel with Imaris using parameters empirically determined to minimize the need for manual curation. Overlapping cells from two channels were merged, and the assignment of cell classes for each individual cell was based on the ratio of integrated intensity within 10 μm (the cell diameter) around the respective determined centers between two channels. This analysis provides quantification of co-localization to support our observation of anatomical distributions of different cell classes.

## Cell sorting and RNAseq

We developed a fluorescence-activated cell sorting (FACS) strategy to isolate proliferative stem cells. We used DyeCycle Violet (DCV) to label live cells proportionally to their DNA content and sorted replicating cells at either S or $G_2$/M phase (*Hayashi et al., 2006*). Schistosomes are covered by a syncytial outer layer impenetrable to typical digestive enzymes used for cell dissociation (*Hahnel et al., 2013*). To overcome this barrier, we briefly treated the parasites with detergents, followed by trypsin to dissociate tissues into cell suspensions. This method dramatically improved the yield of dissociation and reduced the duration of enzymatic digestion to maximize cell viability. Specifically, in vitro-transformed mother sporocysts were permeabilized in PBS containing 0.1% Triton X-100% and 0.1% NP-40 for 20 s, and washed thoroughly to remove the surfactants. The permeabilized sporocysts were dissociated in 0.125% trypsin in HBSS for 10 min and triturated with a 1 mL pipette for 10 min. Cell suspensions were passed through a 100 μm nylon mesh (Falcon Cell Strainer) and centrifuged at 150 *g* for 5 min. Cell pellets were gently resuspended, passed through a 30 μm nylon mesh, and stained with Vybrant DyeCycle Violet (5 μM, Invitrogen), TOTO-3 (0.2 μM, Invitrogen), and calcein AM (0.1 μg/mL, Invitrogen) in sporocyst culture medium for 30–45 min. Dissociation of juveniles was performed similarly but with the following modifications: juveniles were permeabilized for 30 s, dissociated in 0.25% trypsin for 20 min, and triturated with serially narrowed flamed-tip glass. Dissociated cells were analyzed on an LSR II flow cytometer or sorted using a FACAria II flow sorter (BD Biosciences), with dead cells excluded based on TOTO-3 fluorescence. We confirmed that the FACS signature of the proliferative cells disappeared as early as 2 days after worms received high doses of X-ray irradiation (*Figure 4—figure supplement 1C*). All flow sort profiles were confirmed on at least three biological replicates.

250,000 stem cells from either sporocysts or juveniles were sorted directly into lysis buffer (Qiagen) supplemented with 0.6% 2-mercaptoethanol (Sigma), and total RNA was purified using Qiagen RNeasy mini kit. After DNase treatment and poly(A) selection, stranded RNA-seq libraries were prepared using TruSeq Stranded RNA Sample Prep kit (Illumina), pooled in equimolar concentrations, and sequenced on a HiSeq2500 sequencer (Illumina) to acquire 100- or 160 bp reads with a depth of

40–100 million reads per library. To compare these transcriptomes of purified cell populations to those of whole animals, we also extracted total RNA from approximately 10,000 miracidia, in vitro-transformed mother sporocysts (48 hr post-transformation), or cercariae, or about 1000 juvenile worms, using the standard Trizol (Invitrogen) extraction method. All RNAseq data have been submitted to SRA and are available under accession number PRJNA395457.

## Single-cell RNAseq

Single stem cells from in vitro-transformed mother sporocysts were captured on a medium-sized (10–17 μm) microfluidic RNA-seq chip (Fluidigm) using the Fluidigm C1 system. Sorted cells were resuspended at a density of 300 cells/μL, with size distribution and number density confirmed on a TC20 cell counter (Bio-rad). The single-cell suspension was then mixed with Fluidigm suspension reagent at 7:3 ratio and loaded onto the chip immediately. After capture, chambers on the chip were examined quickly by phase-contrast microscopy to assess the number, size, and morphology of captured cells and by fluorescence microscopy to examine the live-dead cell stain, and only chambers containing single round-shaped live cells were included in the downstream procedures. cDNAs were prepared on the chip using SMARTer Ultra Low RNA kit for Illumina (Clontech) following the manufacturer's instructions. cDNA quality was quantitated by qPCR analysis of two quality-control genes (*ago2-1* and *h2a*) on an Applied Biosystems Step One Plus station using GoTaq qPCR reagents (Promega). Libraries were constructed from this cDNA using Illumina Nextera XT DNA Sample Preparation kit. Library size distribution and concentration were assessed using High Sensitivity DNA analysis kit on an Agilent Bioanalyzer, as well as fluorometrically using Qubit Fluorometer (Invitrogen). Libraries were then sequenced on a HiSeq 2500 to obtain 100 bp reads at a depth of 3–10 million reads per cell. Data from four biological replicates were pooled together for analysis, but two of them were later excluded from final results, as they had a high 'dead cell' rate.

Mapping of reads to the annotated *S. mansoni* genome (**Protasio et al., 2012**) was performed using CLC Genomics Workbench 7.0 (CLC Bio) using standard settings. Expression levels were estimated based on transcripts per million (TPM). Ratio of expression levels between libraries was calculated using TPM+1, and TPM values below or equal to one were considered as being dominated by noise (**Kumar et al., 2014**). In a subset of cells that passed quality control based on live-dead stains, quantitative PCR (qPCR) failed to detect *ago2-1* and *h2a*, two ubiquitous stem cell markers (**Wang et al., 2013**; **Collins et al., 2013**). RNAseq on these cells exhibited lower overall genomic mapping rates (<60%), and detected significantly fewer transcripts per cell, more intergenic reads, and strong amplification bias towards the most abundant housekeeping genes. These cells were excluded from downstream analysis. The remaining libraries were then inspected for uniformity in mapping depth throughout the *ago2-1* transcript and the outlier cells were further excluded, as the schistosome stem cells (many of them were dividing) may be sensitive to the microfluidic capture process, leading to mRNA degradation. Although this stringent data selection step reduced the number of cells analyzed, it suppressed variations between cells that are likely attributable to technical artifacts.

PCA was performed using genes expressed in at least three cells and showing variance and bimodality coefficient of expression levels across all cells greater than empirically determined thresholds. Hierarchical clustering was performed using Euclidean distance metric on expression levels standardized gene-by-gene by mean-centering and dividing by the standard deviation of expressing cells. Assignment of cell classes is based on hierarchical clustering. As TPM values are well characterized as log-normal distributions for housekeeping genes, $\log_2(TPM+1)$ was used as a measure of expression level in PCA and hierarchical clustering.

## Single-cell qRT-PCR

For single-cell analysis of juvenile stem cells, cells were sorted directly into 96-well plates that contained 5 μL 1X CellsDirect One-Step Reaction mix (Invitrogen) supplemented with 0.05 μL RNase-OUT (Invitrogen) in each well. Sorted cells were immediately frozen on dry ice and kept at −80°C until reverse transcription (RT). After thawing plates containing sorted cells, each well was supplemented with 5 μL 1X CellsDirect One-Step Reaction mix that also contained 0.2 μL SuperScript III/Platinum Taq mix (Invitrogen) and outer primer pairs with a final concentration of 10 nM per primer. Reverse transcription was performed at 50°C for 20 min and stopped by heating the plate to 95°C

for 2 min. The cDNA was then amplified for 20 cycles (95°C 15 s, 60°C 4 min) before digestion of the remaining outer primers with *ExoI* (20 U/well, New England Biolabs) at 37°C for 30 min and inactivation of the RT enzyme at 80°C for 15 min. Amplified cDNA samples were diluted 1:6 in water. We performed 10 technical replicates (each replicate consisting of one 96-well plate).

For quality control, 5 µL of each well was used to quantify *h2a* levels by qPCR. We randomly picked cells from wells that generated $C_T$ values within 4 $C_T$ around the most probable values (~75% of total wells) for multiplex qPCR on the Fluidigm Biomark platform. For each reaction, 5 µL of diluted cDNA was loaded on a 96.96 DynamicArray IFC chip (Fluidigm) along with negative controls. Expression levels were assessed using inner (nested) primers for each gene. The primer sets are listed in *Supplementary file 2*.

$C_T$ values from the DynamicArray chip qPCR were determined from amplification curves with Fluidigm Real-time PCR analysis software using auto (detectors) thresholding and linear (derivative) baseline correction with a quality threshold of 0.65. The limit of detection was determined as 22 $C_T$ based on negative controls; undetected genes or $C_T$ values greater than 22 were all adjusted to 22. Expression values in log space were calculated as 22-$C_T$. About 10% of cells showing substantially fewer numbers of genes detected were excluded from downstream analysis. To estimate technical variability, two independent sets of nested primers were designed for three genes, *ago2-1*, *mier* (Smp_101370), and *hmt* (Smp_055310), expression levels of which cover the full dynamic range of the qPCR analysis. The technical noise was determined as 2–3 $C_T$ and inversely correlated with gene expression level. PCA was performed on 90 amplicons (87 genes with the extra three technical-variability controls). Subsequently, genes with the highest scores in the first two PCs were identified and used for hierarchical clustering of cells. For hierarchical clustering, expression levels were standardized gene-by-gene by mean-centering and dividing by the standard deviation of expressing cells.

### *In toto* imaging of schistosome-infected snails

Fixed and bleached whole snails were rendered transparent by clearing in 50, 75, and 100% tetrahydrofuran (THF) in water, followed by dichloromethane, and hexane, successively, for 12–24 hr at each step (*Ertürk et al., 2012*). Specimens were then rehydrated through 100% THF, 50% THF, and PBSTx (PBS with 0.3% Triton X-100), and then bleached either in 6% $H_2O_2$ in PBSTx overnight or in 0.5% formamide, 0.5X SSC, and 1.2% $H_2O_2$ for 90 min. Lectin stainings were performed as previously described (*Wang et al., 2013*), with 12–24 hr incubation times at every step. Imaging was performed in RapiClear (Sunjinlab).

## Acknowledgements

*S. mansoni* was provided by the NIAID Schistosomiasis Resource Center for distribution through BEI Resources, NIH-NIAID Contract HHSN272201000005I. We thank J Gao, T Chong, S R Bray and J Li for experimental assistance, and R Roberts-Galbraith and T Rozario for critical reading of the manuscript. Flow cytometry and sequencing were performed at the W M Keck Biotechnology Center at UIUC with technical expertise from M Band, B Pilas, and A Hernandez. BW is supported by a CASI award from the Burroughs Wellcome Fund and a Beckman Young Investigator Award. PAN is an investigator of the Howard Hughes Medical Institute.

## Additional information

### Competing interests

Phillip A Newmark: Member of eLife Board of Reviewing Editors. The other authors declare that no competing interests exist.

### Funding

| Funder | Grant reference number | Author |
| --- | --- | --- |
| Burroughs Wellcome Fund | Career Award at the Scientific Interface | Bo Wang |

| Arnold and Mabel Beckman Foundation | Young Investigator Award | Bo Wang |
| Howard Hughes Medical Institute | Investigator Award | Phillip A Newmark |

The funders had no role in study design, data collection and interpretation, or the decision to submit the work for publication.

### Author contributions

Bo Wang, Conceptualization, Supervision, Funding acquisition, Investigation, Methodology, Writing—original draft, Writing—review and editing; Jayhun Lee, Investigation, Methodology, Writing—review and editing; Pengyang Li, Investigation, Writing—review and editing; Amir Saberi, Investigation, Methodology, Writing—original draft; Huiying Yang, Investigation; Chang Liu, Formal analysis; Minglei Zhao, Formal analysis, Supervision; Phillip A Newmark, Conceptualization, Supervision, Funding acquisition, Writing—original draft, Writing—review and editing

### Author ORCIDs

Bo Wang http://orcid.org/0000-0001-8880-1432
Phillip A Newmark http://orcid.org/0000-0003-0793-022X

### Ethics

Animal experimentation: In adherence to the Animal Welfare Act and the Public Health Service Policy on Humane Care and Use of Laboratory Animals, all experiments with and care of vertebrate animals were performed in accordance with protocols approved by the Institutional Animal Care and Use Committee (IACUC) of the University of Illinois at Urbana-Champaign (protocol approval number 15134), the University of Wisconsin-Madison (protocol approval number M005569), and Stanford University (protocol approval number APLAC-30366).

### Decision letter and Author response

Decision letter https://doi.org/10.7554/eLife.35449.028
Author response https://doi.org/10.7554/eLife.35449.029

## Additional files

### Supplementary files

• Supplementary file 1. Sporocyst stem cell class-dependent genes and their mean expression levels in each stem cell class. Genes exhibit significant differences (>4 fold, $p < 0.01$) between δ, κ, and φ stem cell classes in sporocysts. p-values were calculated by a two-sample t-test, using the more stringent estimate from the assumptions of either unequal or equal population variances.
DOI: https://doi.org/10.7554/eLife.35449.019

• Supplementary file 2. Nested primer sets used for single-cell qPCR. Feature IDs are given in the *S. mansoni* genome version 5 (*Protasio et al., 2012*), but all sequences input for primer design were manually curated by pooling all our RNAseq reads together and mapping to the reference genome. All primer sets were validated by bulk qPCR on whole animal cDNA libraries. Note that for three targets, Smp_179320, Smp_101370, and Smp_055310, we have two independent nested primer sets.
DOI: https://doi.org/10.7554/eLife.35449.020

• Supplementary file 3. Gene names used in the main text and figures. Genes are associated with feature IDs in the *S. mansoni* genome version 5 (*Protasio et al., 2012*). Their functional annotations and cloning primer sequences are also listed.
DOI: https://doi.org/10.7554/eLife.35449.021

• Transparent reporting form
DOI: https://doi.org/10.7554/eLife.35449.022

## Data availability

All RNAseq data have been submitted to SRA and are available under accession number PRJNA395457.

The following dataset was generated:

| Author(s) | Year | Dataset title | Dataset URL | Database, license, and accessibility information |
|---|---|---|---|---|
| Wang B, Saberi A, Newmark PA | 2017 | Single-cell analysis of stem cells driving the parasitic life cycle of *Schistosoma* | https://www.ncbi.nlm.nih.gov/bioproject/395457 | PRJNA395457 |

The following previously published dataset was used:

| Author(s) | Year | Dataset title | Dataset URL | Database, license, and accessibility information |
|---|---|---|---|---|
| Wang B, Collins JJ, Newmark PA | 2013 | Functional genomic characterization of germinal cells in larval *Schistosoma mansoni* | https://www.ncbi.nlm.nih.gov/geo/query/acc.cgi?acc=GSE48282 | Publicly available at the NCBI Gene Expression Omnibus (accession no. GSE48282) |

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
