## [Decision Letter]

Thank you for submitting your article "Stem cell heterogeneity drives the parasitic life cycle of *Schistosoma mansoni*" for consideration by *eLife*. Your article has been reviewed by 3 peer reviewers, including Alejandro Sánchez Alvarado as the Reviewing Editor and Reviewer #1, and the evaluation has been overseen by Marianne Bronner as the Senior Editor. The following individuals involved in review of your submission have also agreed to reveal their identity: Peter W Reddien (Reviewer #2); Peter Olson (Reviewer #3).

The reviewers have discussed the reviews with one another and the Reviewing Editor has drafted this decision to help you prepare a revised submission.

Summary:

Wang et al., provide the first detailed single-cell analysis of stem cell heterogeneity in *Schistosoma mansoni*, a parasitic organism of great global clinical importance, across not one but multiple stages of its remarkably complex life cycle.

The authors use extensive transcriptome profiling of multiple and single, proliferating cells, together with ISH, qPCR and functional analyses that substantiate their quantitative results. It represents another major advance by these authors in our understanding of the genetics that underpin development in parasitic flatworms, and an important comparator to planarians, tapeworms and other flatworms – a phylum of 'super regenerators' that have long merited wider investigation.

Most notable perhaps is the discovery and robust evidence provided for the existence of a factor capable of inducing the germline in this parasite. Pinpointing the nature of germ cell origins in this animal as being driven by induction is, in our opinion, a great contribution to our understanding of parasitology.

Essential revisions:

1) Unless we missed it, there is little to no detailed description of the molecular nature of *eled*. For starters, Is *eled* an acronym? If so where in the paper is it defined? The first instance of its occurrence we could find was in the last paragraph of the subsection “Stem cells in juveniles reveal germline and somatic populations”, as "eledh". Later in the Discussion, *eled* is described as being "…specific to the intramammalian stage of the life cycle and defines the final class of schistosome stem cells, ε-cells, identified in this study", which strikes us as a byzantine circular definition of this molecule. Given the importance ascribed to this molecule as an inducer of the germline, more detail would be expected. Is this molecule unique to *S. mansoni* or is it present in other parasites, for example? Is it a single copy gene in the *S. mansoni* genome or are there isoforms/duplicated genes? Are there any recognizable motifs? What type of molecule is it predicted to be, i.e., secreted, transcription factor, membrane associated, etc.? Maybe we should know this, but I figure that if we don't most of the broad readership of *eLife* may share the same puzzlement. Whether there is or there is no specific information available for this gene should be explicitly stated or the reader is left with the impression of an entirely mysterious molecule.

2) Figure 2. The authors used single of expression of *fgfrA* (negative for *nanos-2*) to mark phi cells; similarly expression of *nanos-2* but not *fgfrA* to mark nu cells. The absence of expression of a marker is weaker classification evidence than the presence of a unique marker because absence could be in some instances explained by technical detection limitation. *klf* and *hesl* would in principle, be more specific class markers. If these probes provide weaker signal would they work in case study single channel FISH to corroborate conclusions? Additionally, the presence of a single marker in common in the embryos to the sporocytes doesn't necessarily indicate these cells lack unique stage-specific attributes. For example, the early blastomeres of a planarian embryo express embryo-specific genes. This could be noted in the text.

3) Epsilon cells: We surmise that the argument for a contribution to soma in posterior growth is because of the very abundant presence of these cells in the posterior of males, which lack sexual structures in this region. If so, this could be explicitly stated. Do available methods allow assessment of whether the phenotype of *eled* RNAi animals is consistent with the model that epsilon cells produce posterior somatic cells for substantial posterior growth? Moreover, what happens to *klf* expression in juveniles? Is it expressed in epsilon cells? Also, the model in Figure 7 could list *klf* and *nanos* as expressed in nu cells. Addressing these issues would improve the clarity of the manuscript.

4) Because this manuscript is laying down an important foundation for defining stem cell lineages in *S. mansoni*, we would like to see a much more fleshed out discussion of the lineages proposed in Figure 7. As presented, there is some ambiguity as to the source of the *eled* cells. For instance, the first paragraph of the subsection “Stem cell classes display distinct spatiotemporal patterns throughout asexual development” seems central to aspects of the lineage argument but are not clearly worded, and further discussion evidence from Schutte 74 might help the arguments made. Also, how much expression overlap is there between cell types as they arise during the complex development of this parasite? For instance, are there transcriptional cohorts shared between nu cells and epsilon and δ cells? Or what about δ and δ prime and δ and psi cells? In other words, understanding why the present model is favored over others would strengthen the conclusion of this otherwise fascinating paper.

---

## [Author Response]

1) Unless we missed it, there is little to no detailed description of the molecular nature of eled. For starters, Is eled an acronym? If so where in the paper is it defined? The first instance of its occurrence we could find was in the last paragraph of the subsection “Stem cells in juveniles reveal germline and somatic populations”, as "eledh". Later in the Discussion, eled is described as being "…specific to the intramammalian stage of the life cycle and defines the final class of schistosome stem cells, ε-cells, identified in this study", which strikes us as a byzantine circular definition of this molecule. Given the importance ascribed to this molecule as an inducer of the germline, more detail would be expected. Is this molecule unique to S. mansoni or is it present in other parasites, for example? Is it a single copy gene in the S. mansoni genome or are there isoforms/duplicated genes? Are there any recognizable motifs? What type of molecule is it predicted to be, i.e., secreted, transcription factor, membrane associated, etc.? Maybe we should know this, but I figure that if we don't most of the broad readership of eLife may share the same puzzlement. Whether there is or there is no specific information available for this gene should be explicitly stated or the reader is left with the impression of an entirely mysterious molecule.

We apologize that this information was buried in a supplemental figure legend (original Figure 4—figure supplement 3), where it was, indeed, far too easy to miss. We have now moved this description to the main text:

*"eled* is a single-copy gene (Smp_041540) that was previously annotated as a nuclear hormone receptor, *dhr4* (Protasio et al., 2012; Berriman et al., 2009); however, our analysis suggests that this gene does not encode a hormone receptor (Figure 4—figure supplement 4). […] Homologs of *eled* are found across *Schistosoma* species, whereas planarian and tapeworm genomes appear to lack them."

We have also carried out protein structure analysis that is presented in the new Figure 4D and described in the main text:

“Figure 4D shows the predicted architecture of *eled*, which has a transmembrane domain at the N-terminus (probability ~100%), a putative transmembrane domain at the C-terminus (probability ~35%), and an extracellular domain in between (probability ~85%). The extracellular domain contains 19% serine and 17% threonine, presenting the highest serine/threonine fraction in known proteins.”

2) Figure 2. The authors used single of expression of fgfrA (negative for nanos-2) to mark phi cells; similarly expression of nanos-2 but not fgfrA to mark nu cells. The absence of expression of a marker is weaker classification evidence than the presence of a unique marker because absence could be in some instances explained by technical detection limitation. klf and hesl would in principle, be more specific class markers. If these probes provide weaker signal would they work in case study single channel FISH to corroborate conclusions? Additionally, the presence of a single marker in common in the embryos to the sporocytes doesn't necessarily indicate these cells lack unique stage-specific attributes. For example, the early blastomeres of a planarian embryo express embryo-specific genes. This could be noted in the text.

We agree with the reviewers that corroborating the cell class identities with additional specific markers would be ideal. We have been able to show that a subset of *fgfrA*^+^ cells expresses *hesl* (Figure 1—figure supplement 2), consistent with the distinction between δ and φ cells. However, in spite of repeated attempts to improve the sensitivity of our FISH protocols for the intramolluscan stage, we have been unable to detect *klf* expression during this portion of the life cycle. Additional work will be required to improve the detection sensitivity of our FISH protocols during intramolluscan development. We have added in the main text:

“Unfortunately, the κ class-specific marker klf was expressed at very low levels (Figure 1E), beneath the detection limits of our current FISH protocol.”

The reviewers are correct to point out that these cells may express other stage-specific genes, and we have added this point to the Discussion:

“Furthermore, in the planarian embryo blastomeres produce temporary embryonic tissues but also give rise to neoblasts by downregulating a set of embryo-specific genes and upregulating genes associated with adult development (e.g., *zfp-1* and *p53*) (Davies et al., 2017). Our data suggest that a similar transition may also occur between κ-cells and δ/δ’cells."

3) Epsilon cells: We surmise that the argument for a contribution to soma in posterior growth is because of the very abundant presence of these cells in the posterior of males, which lack sexual structures in this region. If so, this could be explicitly stated. Do available methods allow assessment of whether the phenotype of eled RNAi animals is consistent with the model that epsilon cells produce posterior somatic cells for substantial posterior growth? Moreover, what happens to klf expression in juveniles? Is it expressed in epsilon cells? Also, the model in Figure 7 could list klf and nanos as expressed in nu cells. Addressing these issues would improve the clarity of the manuscript.

We have clarified the text in the last paragraph of the subsection “Stem cells in juveniles reveal germline and somatic populations”, to state that *eled* somatic expression is concentrated in the PGZ, which lacks reproductive organs in males. This pattern is distinct from that of the proliferation marker, *h2a*, which labels all stem cells and is more evenly distributed throughout the worm. Quantification shows that ~50% of the somatic *h2a*+ cells in the PGZ (1679/3576) and ~10% of the somatic *h2a*+ cells outside the PGZ (769/5366) expressed *eled*.

In the effort to experimentally assess cell differentiation in the PGZ after *eled* RNAi, we have performed EdU pulse-chase experiment. However, we noted a global reduction of EdU incorporation after extended in vitro culture (current RNAi methodology takes ~2 weeks for effective knockdown), even in control animals. In this case, the technical difficulty in sustaining schistosome juvenile growth and development in vitro complicated interpretation of RNAi phenotypes, and we are currently in the process of developing new methods to disrupt gene functions in parasites while they are still inside of mouse hosts.

Finally, we have modified Figure 4C to show that *klf* is expressed in both ε- and δ’-cells. We have also modified the model presented in the new Figure 8 to include *klf*.

4) Because this manuscript is laying down an important foundation for defining stem cell lineages in S. mansoni, we would like to see a much more fleshed out discussion of the lineages proposed in Figure 7. As presented, there is some ambiguity as to the source of the eled cells. For instance, the first paragraph of the subsection “Stem cell classes display distinct spatiotemporal patterns throughout asexual development” seems central to aspects of the lineage argument but are not clearly worded, and further discussion evidence from Schutte 74 might help the arguments made. Also, how much expression overlap is there between cell types as they arise during the complex development of this parasite? For instance, are there transcriptional cohorts shared between nu cells and epsilon and δ cells? Or what about δ and δ prime and δ and psi cells? In other words, understanding why the present model is favored over others would strengthen the conclusion of this otherwise fascinating paper.

We have redrawn the new Figure 8 to clarify the model (note that we have changed ν to κ-cells as suggested by the reviewers). Concerning the source of the ε-cells, we have expanded the Discussion to read:

“Based on the antagonistic interaction between *eled* and *nanos*, we suggest that κ-cells downregulate *nanos-2* and activate expression of *eled*. […] The absence of δ/δ’-cell markers, *fgfr, zfp-1, p53*, and *hesl*, in both κ and ε-cells favors the model that κ/ε-cells form a separate lineage from δ/δ’-cells.”

To be more specific when referring to the early histological studies, we have added additional evidence from Schutte 1974 to the Discussion:

“We posit that κ cells serve as “embryonic” stem cells: in addition to *nanos-2* they express a *klf* homolog and are found isolated in developing mother sporocysts. […] These studies reported that the scattered stem cells may persist in mother sporocysts for months and continuously undergo asymmetric divisions, with one daughter cell forming an embryo while the other remains undifferentiated (Schutte, 1974).” And also:

“The distinction between κ, δ, and φ-cells is also consistent with early histological studies that observed germinal cells in three characteristic tissue locations within mother sporocysts: either scattered among daughter embryos, clustered inside of embryos, or situated close to mother sporocyst walls (Schutte, 1974; Pan, 1965).”

Finally, we have added a new figure (Figure 1—figure supplement 3) to quantify expression overlap and differences between κ, δ, and φ-cells. We have added in the main text:

“In addition to these class-defining genes, the divergence of the three cell classes is manifested by hundreds of other genes that exhibit various levels of statistically significant differences between classes (Figure 1—figure supplement 3). […] These observations confirm that sporocyst stem cells, regardless of the subpopulation to which they belong, share a common transcriptomic profile.”

Because we used single-cell qPCR to classify juvenile stem cells, it is impossible to compare cell classes between intramolluscan and intramammalian stages across transcriptomes. However, among the 90 genes assayed by qPCR, only *eled* emerges as a juvenile-specific gene that defines a new cell class. This result is consistent with the transcriptional uniformity observed among sporocyst stem cells.